# Usability and Engagement Testing of mHealth Apps in Paediatric Obesity: A Narrative Review of Current Literature

**DOI:** 10.3390/ijerph19031453

**Published:** 2022-01-27

**Authors:** Niamh Arthurs, Louise Tully, Grace O’Malley, Sarah Browne

**Affiliations:** 1W82GO Child and Adolescent Obesity Service, Children’s Health Ireland at Temple Street, D01 XD99 Dublin, Ireland; graceomalley@rcsi.com; 2School Public Health, Physiotherapy & Sports Science, University College Dublin, D04 V1W8 Dublin, Ireland; sarah.browne1@ucd.ie; 3School of Physiotherapy, Division of Population Health Sciences, RCSI University of Medicine and Health Sciences, D02 YN77 Dublin, Ireland; louiseTully@rcsi.ie

**Keywords:** childhood obesity, mobile health, usability, engagement, testing methods

## Abstract

Mobile health (mHealth) platforms have become increasingly popular for delivering health interventions in recent years and particularly in light of the COVID-19 pandemic. Childhood obesity treatment is an area where mHealth interventions may be useful due to the multidisciplinary nature of interventions and the need for long-term care. Many mHealth apps targeting youth exist but the evidence base underpinning the methods for assessing technical usability, user engagement and user satisfaction of such apps with target end-users or among clinical populations is unclear, including for those aimed at paediatric overweight and obesity management. This review aims to examine the current literature and provide an overview of the scientific methods employed to test usability and engagement with mHealth apps in children and adolescents with obesity. A narrative literature review was undertaken following a systematic search. Four academic databases were searched. Inclusion criteria were studies describing the usability of mHealth interventions for childhood obesity treatment. Following the application of inclusion and exclusion criteria, fifty-nine articles were included for full-text review, and seven studies met the criteria for usability and engagement in a clinical paediatric population with obesity. Six apps were tested for usability and one for engagement in childhood obesity treatment. Sample sizes ranged from 6–1120 participants. The included studies reported several heterogenous measurement instruments, data collection approaches, and outcomes. Recommendations for future research include the standardization and validation of instruments to measure usability and engagement within mHealth studies in this population.

## 1. Introduction

### 1.1. Background to mHealth

The scale and complexity of treating child and adolescent obesity place substantial demands on healthcare resources. Delivery of best-practice multidisciplinary, family-based treatment programmes requires extensive networks of adequately trained healthcare professionals, administration, appropriate clinical treatment spaces, and time [1]. Direct associations exist between treatment intensity, the involvement of parent/caregivers and healthcare outcomes for children [2,3,4]. There is a need, therefore, to ensure intervention intensities are designed to deliver meaningful clinical outcomes. However, attrition rates for paediatric obesity treatment can present challenges to health systems, clinical teams and the provision of adequate resources and funding. Reasons for attrition and therefore insufficient care for young people with obesity include long travel times for regular appointments, frequent absences from school and work and clinical settings that are not easily accessible for some families [5,6,7]. Providing early intervention for paediatric obesity is warranted as obesity can severely impact on the quality of life of the young person and increase their likelihood of having obesity as an adult [8] and developing complications of obesity [9,10], which places further demand on resources in the long-term.

Technology has enabled significant advancements in healthcare and digital health has been relied on during the ongoing COVID-19 pandemic [11]. Digital health is often termed “ehealth” and is defined by The World Health Organisation as “*the cost-effective and secure use of information and communications technologies in support of health and health-related field, including health care services, health surveillance, health literature, and health education, knowledge and research*” [12]. Mobile Health (mHealth) is a subgroup of digital health and comprises the use of mobile devices, including smartphones and tablet devices to support wellbeing and assist in the management of acute or chronic health conditions, by providing access to certain healthcare resources [13]. Alternatives and enhancements to the traditional face to face model of obesity interventions for young individuals and their families need to be considered in order to optimise timely treatment and address issues such as extensive waiting lists and geographic, economic, or socio-cultural inaccessibility described previously [14]. Despite the acknowledgement that technology can enhance care, it must be used with caution to ensure it does not increase disparities through, for example, excluding those with poor internet access, low digital literacy or physical or intellectual impairment [15]. Current clinical guidelines specify that digital health interventions use certified and regulated tools [16]. Due to the mobile nature of mHealth apps, it is likely that they will be used in multiple environments such as school, home, with family or friends and this could potentially enhance the effectiveness of the intervention beyond the efficacy of interventions based on pen-and-paper or desktop devices [17]. Additionally, smartphone apps could have a purposeful role in adherence to treatment through facilitating behaviour change techniques such as self-monitoring and feedback. Collecting ecological momentary assessment data; that is information gathered in real-time in an individual’s natural environment reduces the incidence of recall bias and can act as additional support to intervention components [13]. According to the most recent app store metrics, there are currently 170,897 apps in the iTunes app store [18] and 95,788 apps in the Google Play store [19] that are classified under “Health and Fitness”. Despite this vast and continually expanding array of health apps, there is not an equivalent contribution to the testing of these apps reported in the clinical literature [20]. Even though many potential benefits of mHealth are widely reported by the media, industry and research communities the evidence for such benefits are not borne out in published scientific evidence. Furthermore, the use of mHealth interventions for treating paediatric conditions such as obesity is an emerging area. This is evident from a recent systematic scoping review on mHealth for paediatric weight management, which identified few studies (*n* = 3) that formally reported usability assessments of the mHealth intervention [21].

### 1.2. mHealth Interventions in Paediatric Obesity Treatment

The complexity of obesity with its mixed aetiology and multiple associated comorbidities complicates the designing and testing of treatment interventions, particularly in deciding which outcome criteria will determine the success of the intervention [14]. The National Institute for Health and Clinical Excellence (NICE) [16] guidelines state that from approximately age 12, young people in weight management programmes should be encouraged to monitor their food intake, physical activity (PA) and sedentary behaviour. Common app features include goal setting [22,23,24,25], peer support [25,26,27], and self-monitoring functions for: activity [23,24,28], diet [27,29] and both [25,30,31,32]. International evidence for paediatric obesity treatment interventions recommends the inclusion of components that target both physical activities (PA) and food/eating behaviours simultaneously [16,33,34]. In a pilot study among adolescents comparing self-monitoring of PA and diet recorded on an app to a paper diary, greater self-monitoring and reduced screen time post-intervention was observed in those using the app compared to the diary [35]. Additionally, those using the app stated that it motivated them to make more healthy food decisions and to exercise [35]. In contrast, another pilot study found low self-monitoring adherence of diet and PA in adolescents who were overweight/obese using a smartphone app and suggested this could be due to the “tedious” and difficult operation of the app [30]. Apps may need more interactive components such as games and social media to improve the attention and ongoing engagement of young people [17,36]. Although one pilot study purposely excluded games, social media and text messaging services, in order to deter app usage beyond the intervention (including for goal-setting and self-monitoring) and reduce rather than encourage screen time [37]. In this same study, the use of the app declined throughout the 12-week intervention, despite the inclusion of a reward system [37]. Other researchers have found utilisation of peer support features in a weight loss app to be negligible, as participants felt uncomfortable interacting with unknown others, especially regarding their weight [27].

Diet and PA are multifaceted behaviours, affected by many psychological, social, and environmental influences [23]. Specific dietary behaviours targeted by mHealth interventions such as increasing fruit and vegetable consumption and decreasing sugar-sweetened beverage intake show mixed results. In one study involving children at high risk of obesity, the mobile technology intervention had small to moderate effects on increasing fruit and vegetable intake and reduced consumption of sugar-sweetened beverages; compared to the control group [37]. However, these differences were not statistically significant and the intervention was only 12 weeks [37]. Conversely, in another study involving children who were overweight or had obesity and a follow-up period of 6 months, medium to large effect sizes and significant improvements on these same outcome measures were observed [22]. This suggests that longer intervention periods are needed in order to observe effective dietary behaviour change in youth with obesity. Behaviour change outcomes that were positively associated with the self-monitoring function of mHealth apps in young people include better self-monitoring of diet and exercise goals [30,38], and the reminder feature of an app contributed to greater PA levels and dietary improvements [23]. These outcomes over time could contribute to healthier BMIs [39].

### 1.3. Technical Usability and Engagement Testing

Considerable testing of apps by end-users is required to reveal the best methods and design of the app for particular clinical groups [39] and enable the research team to discover and solve any technical glitches that could profoundly impact the utility of the app [23]. The first stage of assessing mHealth interventions is piloting and usability testing [40], as this is a key determinant of their use and approval [41], prior to the subsequent stages of assessing the efficacy and disseminating findings. User engagement involves the quality of the user experience, outlooks and expressions of their interactions, and their want or need to use the app for longer amounts of time or continually [42]. The development of effective mHealth interventions relies on understanding user engagement yet engagement requires quite intricate metrics due to its multifaceted, complex nature and influences that can impact its use including family, community, culture and context [43]. Approaches to measuring user engagement are manifold [44] but those used for testing mHealth interventions amongst children and adolescents with obesity are not widely known. Furthermore, there is ambiguity in the literature around definitions of usability and user experiences in digital health studies, and these can differ between fields of study and researchers [45].

Usability is the degree to which a product can achieve specific tasks efficiently, effectively and satisfactorily by identified users and in an established setting [46]. The following measures are uniformly used in testing usability: efficiency (the apparent capacity of the system to finish tasks in a competent, effective and rational way), affect (the user’s emotive responses for the system), helpfulness (the compatibility of the system in helping to solve challenges), controllability (the perception that the system is constantly interacting with the input of users), and learnability (the belief that the system is fairly easy to familiarise with) [47]. Qualitative research including interviews and observation methods can be optimal ways of collecting detail on the experiences, interactions and responses of users which cannot be evaluated using other methods such as surveys or system logs [48]. Despite this, the majority of research findings convey that insufficient detail is gathered from these techniques [48]; highlighting the benefits of a mixed-methods approach using both qualitative and quantitative processes.

Given the requirement for mHealth solutions for paediatric obesity treatments and the fast-emerging field of mHealth interventions in this population, there is a need for usability and engagement measurement tools for use in research and practice. We aimed to summarise the published literature describing studies that assessed usability and/or engagement of mHealth interventions for childhood obesity treatment and their findings.

## 2. Methods

A narrative literature review was conducted, after implementing a systematic search (Appendix A) in order to capture the breadth of published work and synthesise study characteristics including methods used and findings. For increased validity, two independent reviewers conducted title, abstract and full-text screening.

### 2.1. Information Sources and Search Strategy

A formalised search of four electronic databases including the Cochrane Library, Web of Science, Scopus, and PubMed was undertaken on 12 August 2021. Literature published from 2009 onwards (up to August 2021) was included as smartphones were first brought to mainstream market with the introduction of the Apple iPhone in 2007 followed by Android models including Samsung and High Tech Computer Corporation (HTC) in 2008 [49] and it is unlikely that mobile apps were widely available before this. Table 1 describes the population, concept and context (PCC) defined to help develop the search protocol and key search terms.

Key search terms developed from PCC outlined in Table 1 included combinations and variations of: “children, childhood, adolescents, young people”, “obesity”, “usability testing”, “usability evaluation”, “user engagement”, “user satisfaction”, “mHealth apps”, “mobile health”, “mobile applications”, “mHealth tools”, “mobile technology”. The full search strategy is available to view in Appendix A.

### 2.2. Study Selection

#### 2.2.1. Inclusion Criteria and Searches

Appendix B outlines the search strategy used which resulted in 59 identified articles for full-text review (Figure A1) and resulted in seven final included articles which met all of the following inclusion criteria:Written in English,Involving children and/or adolescents with overweight or obesity (as defined by the study authors),Under the age of 18 years,Focussed on usability and engagement testing of mHealth applications.

Interventions involving participants older than 18 years or those based on text messages or other digital health platforms apart from mHealth apps were excluded.

#### 2.2.2. Outcome Measures

The primary outcome of interest was measures of overall usability and engagement as reported by original studies. Given the limited and novel research for mHealth usability, no restrictions on methods of measurement were applied. Secondary outcomes of interest included technical usability and user satisfaction.

## 3. Results

### 3.1. Characteristics of Included Studies

As illustrated in Table 2 below, seven studies met the inclusion criteria in terms of reporting usability and/or engagement testing among a paediatric population receiving treatment for obesity. Three studies recruited participants who were receiving obesity treatment in paediatric hospitals [24,32,50]. One study recruited adolescent participants from an evidence-based and medically led weight-loss summer camp including participants from backgrounds of poverty and housing insecurity across 50 states and 23 countries in America [26]. One article involved families attending primary care centres for children with overweight issues in Italy [51]. Another article selected families from purposive sampling techniques using enlisted paediatricians of a local research institute in Italy [52]. One study involved a commercial, mHealth app that was either paid for by the participants’ parents, the parents’ employers or a family health insurance plan and this level of cover influenced the commitment period assigned from 4 to 24 weeks [31]. The mHealth app studies identified targeted behaviours around PA alone (*n* = 1) or both diet and PA (*n* = 6). The apps included functions for self-monitoring (*n* = 7), tips and advice (*n* = 5), social support (*n* = 3), goal setting (*n* = 2), and rewards (*n* = 1) as depicted in Table 2 below. The majority (86%) of these studies employed both quantitative and qualitative techniques to test usability (*n* = 6) and engagement (*n* = 2). One study (*n* = 1) used quantitative methods only and measured engagement alone, not usability.

Table 2 summarises the identified studies (*n* = 7) that tested usability and engagement with apps aimed at children/adolescents with overweight/obesity specifically, including the directly relevant study characteristics.

### 3.2. Methods for Evaluating Usability and Engagement in Current Literature

Usability was measured across studies using various usability definitions and methods. One article stated that usability is not fixed as it is only relevant to the specific context in which it is applied [52]. Another study specifically measured “technical usability” by evaluating the relative user efficiency of an app; which compares task completion time between experts and novice end-users, the latter was a group of adolescents living with obesity [24]. Questionnaires were also used to test usability and were successfully tailored for youth and adapted for evaluating smartphone apps including the System Usability Scale (SUS) [32,52] and the standardised software usability measurement inventory (SUMI) [24]. Both of these questionnaires were completed following the task phase of testing and each were combined with open-ended questions for qualitative feedback on the apps [24,32]. SUMI was used to measure overall user satisfaction with an mHealth app, however, authors recommended further testing of each app feature and measuring satisfaction on completion of each task in future studies [24]. SUS was used to measure perceived usability of various mHealth apps in youth study populations [52]. One study incorporated a “*SUS ideal format*” to compare usability scores for two different versions of the same app [52]. Another article purposely used a seven-point Likert scale composed of single-item measures in order to reduce the burden on their study population of children [50]. The majority of studies have analysed the quantitative data from these questionnaires using descriptive statistics [24,32,50,51]. Qualitative data from focus groups or interviews were commonly analysed using thematic analysis [51]. This method assigns codes to units of meaning in a dataset which can enable researchers to dissect the data and highlight main themes that have the potential to improve an mHealth app [51]. One study that measured participant engagement only as its primary outcome assessed this by the overall collective number of individual coaching sessions received by participants throughout a defined participation period [31]. It also evaluated engagement from the cumulative time that participants interacted with the app, programme retention rates, data logged by participants for food and PA and the collective number of interactions from each participant to their assigned health coach [31].

## 4. Discussion

This study aimed to provide a summary of usability and engagement testing methods of mHealth interventions for childhood obesity treatment. Ideally, to evaluate usability completely, the relationship between users, devices, and tasks in a defined setting needs to be measured [48]. Our findings show that literature describing appropriate methods of testing usability and user engagement with young people with overweight/obesity is limited. Of the studies identified in this review (*n* = 7), a mixed-methods approach provided more contextual findings compared to quantitative alone. This includes focus groups and interviews for qualitative data, and questionnaires and engagement metrics derived from the technology itself for quantitative data. Descriptive statistics were the main method reported for analysing quantitative data and thematic analysis for qualitative data. Future research needs to standardise and validate usability tools with this population. There is also the need for consensus on clear definitions around aspects of usability, for which work is ongoing [45]. A systematic scoping review was conducted on heterogeneous methods of evaluating mHealth for paediatric weight management, however, the search was complete up to January 2019 and few studies had looked at usability [21]. This exploratory review includes an updated search incorporating time during the COVID-19 pandemic, during which digital health was vastly and rapidly deployed in health services across the world. Despite this increased uptake, we still found few studies in this area testing usability and engagement of mHealth interventions in paediatric overweight/obesity treatment. Considering the views and experiences of end-users is fundamental in the digital design process and vital for developing improved iterations to enhance greater use and ultimately augment the value of the digital intervention. The development of a usability checklist for paediatric digital interventions would be a valuable contribution to the field of mHealth research and practice. 

### 4.1. Comparison with Usability and Engagement Testing in Other Areas of Paediatrics

Although there were limited search results specific to paediatric overweight and obesity, the broader search identified further methods of testing usability and engagement of mHealth interventions for other paediatric health conditions including concussion [53] and mild traumatic brain injury [54]. The most extensively used measure of usability function is task success; it is highly probable that issues exist with a digital tool if the user is unable to complete a given task [46]. One method of measuring efficiency or the level of effort that users provide to achieve tasks is by assessing task completion time, including error and correction time [46]. Tasks completed in less time indicate that less effort was required to complete them, which is believed to enhance the overall experience [53]. However, the length of time required will depend on the nature of the task which means time comparisons and efficiency can only be evaluated if the same task is performed [55]. This form of usability testing is useful for identifying possible usability gaps between novice users and expert users [55]. More recent usability testing with youth employed the “think aloud” method, which gathers information on both usability and content of apps as participants engage with them [53,54]. Applying “think aloud” during usability testing of an app with adolescents revealed specific barriers in one study and enabled researchers to adapt the app, including increasing the text size and making it more intuitive [54].

Questionnaires were used in many studies to measure usability, user engagement and satisfaction amongst young people with mHealth apps in the studies identified. Although a major challenge mentioned across the literature is that traditional usability evaluation methods such as surveys and qualitative approaches, are aimed at adults and need to be adapted to account for developmental differences in children and adolescents in addition to privacy and ethical considerations [56]. The Usefulness, Satisfaction, and Ease of Use questionnaire (USE) was used with a youth population to develop semi-structured questions for guiding focus groups [57]. Furthermore, the short and simple nature of the SUS enhances its appropriateness for paediatric populations [32]. Further advantages reported for the SUS include its suitability with small sample sizes, the option to provide further detail and the ability to compare and interpret final scores with reputable standardised benchmarks [32].

### 4.2. Implications for Practice and Research

Although some evidence exists for the effectiveness of eHealth interventions in treating paediatric overweight and obesity, knowledge gaps still remain for determining the most suitable and efficacious intervention features, long-term effects, and sustainability [58]. Moreover, a recently published systematic review of mHealth applications found inconclusive evidence to support the effectiveness of apps for child and adolescent weight management and concluded that this research topic remains in its infancy [59]. mHealth is very heterogeneous as it can comprise of a wide range of features from smartphone applications to text messages, reminders, games or peer support and advice functions. Further research is needed to assess what format, mode of delivery, and combination of parent involvement and face-to-face care integrated with technology are most efficacious. Some apps have shown promising effects on clinical outcomes including measures of anthropometry [27,60,61] and goal setting [37]. Others have not demonstrated any benefit [35]. Notably, for these positive outcomes shown, other factors could have contributed to their success considering that control groups were not included and the low quality that these studies received on the Jadad scale [59]. One paper reported that reductions in zBMI from a previous part of the study, involving both a group and smartphone intervention were not maintained in the second part of the study, involving the digital-only intervention [30]. Another study found that using smartphones solely to decrease BMI in children with obesity has low efficacy [29], and other findings report no significant differences on BMI, waist circumference, or percent body fat [35,36,37].

It is worth noting findings in the literature that investigated the potential benefits and drawbacks of incorporating a combined approach of mHealth and usual care. A recent study evaluated the usage and user experiences of an mHealth support system for paediatric obesity treatment, which was provided alongside standardised care and compared this to standardised care alone [28]. The researchers concluded that the combined approach with the mHealth intervention achieved better clinical outcomes versus the standardised care on its own. However, the authors also advised that these treatment outcomes require a longer study duration using greater sample sizes and further follow-up [28]. In contrast, a different study found higher attrition rates in the mHealth intervention group which included usual care (5/8, 63%), compared to a control group (3/12, 25%) that received usual care alone [32]. The authors suggested that the high attrition may have been influenced by the inferred burden of tasks and behavioural issues in this clinical cohort [32].

Additionally, data protection and privacy policies are major considerations when balancing the relevant potential benefits of using mHealth interventions, especially in this vulnerable group and crucially because such protocols vary extensively worldwide [62]. Furthermore, mHealth interventions are not suitable for everyone and widespread implementation could lead to access issues and increased health inequalities [63]. Any benefits of incorporating mHealth must be balanced with alternative options for those with additional needs such as learning difficulties, reduced dexterity or those from lower socio-economic backgrounds. Considering this, it is essential that implementation of mHealth interventions in paediatric health services is carried out vigilantly based on prior testing, thorough evaluation and evidence for best practice.

### 4.3. Limitations of Current Usability and Engagement Testing Methods

Limitations of usability testing in original studies with young people include response bias with questionnaires and social acceptability bias when survey facilitators are present [56]. However, having a researcher present during the survey may be important to explain the terms used and answer questions, particularly for children with various literacy levels [56]. Simplifying questions and probes used during testing could account for discrepancies in literacy or understanding amongst young participants involved but this could also reduce the quality of feedback [56]. In addition, themes developed from data analysis may be shaped by participation bias in focus groups and interviews [57]. Researchers of an mHealth app have also documented selection bias (variances among those who decide to partake and those who do not) and coverage bias (variations among those who are capable of participating and those who are not) [64]. These researchers are currently working on solving these types of errors by establishing suitable sample weighting to counteract these biases [64].

### 4.4. Limitations of This Review

As this is a narrative literature review, findings are limited and the quality of the included studies has not been evaluated with a quality assessment tool. Conducting a systematic review on this topic following the PRISMA guidelines and assessing the quality of the included studies using an approved quality assessment tool is recommended to further inform the evidence base and those interested in this field, as usability methods and technologies rapidly develop.

## 5. Conclusions

mHealth technology could present a nascent therapeutic option for childhood obesity and the development of mobile apps for paediatric health is increasing. It is difficult to ascertain the direct effect of mHealth technologies on clinical outcome measures for paediatric obesity, including BMI and changes in PA, sedentary activity and eating behaviours, as many studies involved other interventions in addition to mHealth apps.

From the literature search, only six apps aimed at targeting childhood obesity were tested for their usability and one for engagement levels alone; six focussed on diet and all seven incorporated PA functions. In order to advance mHealth apps to a level that they can be used effectively and more extensively for paediatric obesity, further research needs to be channelled into identifying the best practice for technical usability testing and developing evidence-based and expert-led interventions for youth populations.

## Figures and Tables

**Table 1 ijerph-19-01453-t001:** PCC outline to develop search protocol for literature review.

P	Population	Children and adolescents (0–18 years) living with overweight/obesity (as defined by individual studies)
C	Concept	Usability (the extent that a product can complete certain tasks in an effective, efficient and satisfactory manner by specific users and in a defined setting) and engagement as defined by the study authors.
C	Context	mHealth interventions (the use of smart mobile devices such as phones or tablet PCs) to deliver partial or full weight management programmes

**Table 2 ijerph-19-01453-t002:** Summary of studies testing usability and engagement of mHealth applications in children and adolescents with overweight/obesity (*n* = 7).

Reference and Country (City)	Study Design	Sample Characteristics (Age and Condition)	App Features	Method	Usability and Engagement Outcomes Reported
(O’ Malley et al., 2014) [24] Ireland (Dublin)	Quantitative and qualitative	12–17 years living with overweight/obesity (BMI ≥ 98th percentile) (*n* = 10). Female (*n* = 3), male (*n* = 7).	Self-monitoring of PA and food intake, goal setting, social support, tips and rewards.	Time-on-task of novice and expert users. Standardised software usability measurement inventory (SUMI).	Technical usability by end-users. Relative user efficiency score.
(Gabrielli et al., 2017) [51] Italy (Trento)	Quantitative and qualitative	7–12 years who are overweight (BMI 85th–94th percentile) (*n* = 6). Their parents (*n* = 6).	PA and diet monitoring. Diet advice.	System Usability Scale (SUS) questionnaire with parents only. Semi-structured interviews with parents and children.	Usability by end-users and suggestions for further improvements.
(Kowatsch et al., 2017) [50] Switzerland (St. Gallen)	Qualitative and quantitative	Children undergoing treatment for obesity. Female (*n* = 8), male (*n* = 3). Mean age = 12.6 years, SD = 2.4 (*n* = 11).	Self-monitoring of PA.	Observation. Questionnaire. 7-point Likert scale.	Usability and acceptability by end-users.
(Cueto et al., 2019) [31] USA (California)	Quantitative	5–18 years living with overweight (BMI ≥ 85th percentile)/obesity (BMI ≥ 98th percentile) (*n* = 1120).	Self-monitoring of eating and PA. Individualised coaching sessions.	The overall collective number of individual coaching sessions, coaching messages, dietary events, and physical-activity events that participants took part in throughout the participation phase. Additionally, the duration of the participation period and programme retention.	High participant engagement.
(LeRouge et al., 2019) [26] USA (Washington)	Qualitative and quantitative	9–18 years with overweight/obesity (BMI 85th–99th percentile range) Phase 1 (*n* = 48), parents (*n* = 15), HCPs (*n* = 6). Phase 2 12–17 years (*n* = 70), HCPs (*n* = 10).	Social networking, motivation, “recipe builder”, PA and food management.	Think out loud, semantic differential scale and semi-structured interviews.	Usability by end users.
(Browne et al., 2020) [32] Ireland (Dublin)	Qualitative and quantitative	9–16 years with obesity (BMI ≥ 98th percentile) (*n* = 20).	PA and diet monitoring.	System usability score surveys, verbal feedback. Engagement was measured from the number of training meals completed and volume of data collected.	Usability was reported for the BigO app but not for the Mandolean app. Low engagement levels and poor acceptability were reported for both apps. Further technical usability testing was advised.
(Rahman et al., 2020) [52] Italy (Trentino)	Qualitative and quantitative	6–12 Years with obesity (*n* = 6), parents (*n* = 6),	PA and diet monitoring. Nutrition and portion size information.	System usability scale questionnaire and semi-structured interviews focussed on interface preferences, eating behaviours, and user experience.	Both usability and user satisfaction were reported for both versions of the app but appB received a higher average usability score (score > 92) than appA (score > 85) for its friendly interface and elaborative components. Technological modifications suggested.

BMI, body mass index; HCPs, healthcare professionals; PA, physical activity; SD, standard deviation.

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
