# Peer review of "Usability and Engagement Testing of mHealth Apps in Paediatric Obesity: A Narrative Review of Current Literature"

_ijerph, 2022, doi:10.3390/ijerph19031453_

Round 1

Reviewer 1 Report

In my opinion, the article presents an interesting topic that is on the rise. The conceptualization of the topics is clearly worked out. However, the manuscript presents some defects that need to be corrected so that it can be published. 

In general, the article should correct aspects of the structure, especially in Materials and Methods. In addition, I consider that it would be beneficial to proofreading English to ensure correctness, accuracy and clarity.

INTRODUCTION

The authors correctly describe the background and the reasons that lead them to carry out the present study. Just as a recommendation, it would be appropriate to introduce the concepts of digital literacy and eHealth as a basis for its main concept of mHealth.

DATA AND METHODOLOGY

In this area there is a need for improvement, but only in certain details. One of them is when the authors comment “please see Appendix 1”. Only the following should be included Appendix 1.

On the other hand, in Table 1, it is unnecessary to include the Date searched column, due to the dates are the same in all cases. Thus, it would be advisable to mention it in the corresponding section within the text of the study.

RESULTS

They are well organized and clearly visible.

DISCUSSION AND CONCLUSIONS

These sections present a great structure and organization. It is very clear and shows a strong research study.

Author Response

Please see the attachment. Thank you for your comments and feedback. 

Reviewer 2 Report

The research idea of this paper is very good, but the paper needs to be further improved;

(1) For keywords, "youth; mHealth; Interventions" are inappropriate, and mHealth and mobile health are duplicate;

(2) In the "technical usability and engagement testing" part of the introduction, the author mentioned "the first stage of evaluating mHealth interventions is usability testing", but we didn't see the second stage and the definition and expression of engagement testing;

(3) I think the retrieval strategy of identifying only 7 studies from 982 089 literatures is complex and not repeatable, which it is far from enough to summary a review from only 7 studies.

(4) In Appendix A, what does "not peer review" mean?

(5) Whether mHealth apps can improve paediatric obesity, what functions the author thinks the most ideal mHealth apps should achieve, and what suggestions should be made to the existing mHealth apps.

Author Response

(The authors gave the same response as above.)

Reviewer 3 Report

Useful review. Good research. 

Author Response

(The authors gave the same response as above.)

Round 2

Reviewer 2 Report

Useful review. Good research